# Development of a Donkey Grimace Scale to Recognize Pain in Donkeys (*Equus asinus*) Post Castration

**DOI:** 10.3390/ani10081411

**Published:** 2020-08-13

**Authors:** Emma K. Orth, Francisco J. Navas González, Carlos Iglesias Pastrana, Jeannine M. Berger, Sarah S. le Jeune, Eric W. Davis, Amy K. McLean

**Affiliations:** 1Department of Animal Biology, University of California Davis, Davis, CA 95617, USA; emma.kira.orth@gmail.com; 2Department of Genetics, Faculty of Veterinary Sciences, University of Córdoba, 14071 Córdoba, Spain; ciglesiaspastrana@gmail.com; 3The Worldwide Donkey Breeds Project, University of Córdoba, 14071 Córdoba, Spain; 4San Francisco SPCA, Society for the Prevention of Cruelty to Animals, San Francisco, CA 94115, USA; drb@sacvetbehavior.com; 5Department of Surgical and Radiological Sciences, School of Veterinary Medicine, University of California Davis, Davis, CA 95617, USA; sslejeune@ucdavis.edu; 6International Animal Welfare Training Institute, School of Veterinary Medicine, University of California Davis, Davis, CA 95617, USA; ewdavis@ucdavis.edu; 7Department of Animal Science, University of California Davis, Davis, CA 95617, USA

**Keywords:** donkey, *Equus asinus*, facial expression, pain, grimace scale

## Abstract

**Simple Summary:**

Donkeys originally evolved as a desert animal, and unlike the horse (which flees or runs away from danger), the donkey fights to avert danger. Hence, donkeys are more stoic and tend to express fear, pain, and discomfort in more subtle ways than horses. For owners and practitioners, it can prove to be challenging to identify donkeys in a state of pain or discomfort until the animal has reached an advanced degree of disease, at which point veterinary intervention may be too late. This study aims to identify signs of pain from both facial and body parameters in donkeys undergoing a surgical procedure. Scores were based on noted signs of discomfort/pain from the observed body language of the face, such as the eyes, ears, nose, nostrils, and muzzle, along with their overall body appearance. The study demonstrated that developing a scoring system donkey grimace scale proved to be accurate in identifying discomfort related to pain. However, the accuracy of the scale can be influenced by the observers’ gender, level of donkey knowledge, and experience.

**Abstract:**

The objectives of this study were to establish a donkey ethogram, followed by a donkey grimace scale to be applied to donkeys pre- and post-castration and to test if there was a notable difference in scores based on observer knowledge, gender, and experience, which could reveal possible discomfort/pain. Nine healthy male adult donkeys were surgically castrated. Fifty-four photos were selected from frontal, lateral, and body views taken pre- and post-castration. Observers ranging from minimal to extensive knowledge and levels of experience based on education and hours/month spent with donkeys scored six photos/donkey on a scale of 0–2 (0 = not present, 1 = moderately present, 2 = obviously present). Scores were based on body language and facial parameters: Ears down, ears back, eye white showing, glazed look, orbital tightening, eyes round shape, nostril tension, eyes narrow shape, muzzle tension, and abnormal stance and overall perception of the animal being in pain. Level of experience and knowledge, as well as gender significantly (*p* < 0.001), affected observers’ ability to accurately score images. The study suggests that the most significant indicators of pain in donkeys are overall appearance and abnormal body stance provided their sensitivity, specificity and accuracy values of 63.18%, 62.07%, and 62.60%, respectively.

## 1. Introduction

The misconception that donkeys do not feel pain to the degree that horses do, by suggesting donkeys have a higher level of pain tolerance, combined with a difficulty to identify indicators of pain in donkeys [1], has likely deprived many donkeys of receiving treatment when ill or injured [2,3]. Although there have been limited studies examining specific indicators of pain in donkeys, few behavioral signs of pain displayed in horses have not been described in donkeys [2,3,4]. Donkeys likely feel pain and yet display little signs of pain, but currently, there is no evidence that donkeys have a different pain tolerance to that of other equines [2,5,6].

Animal pain has been described as “an aversive sensory experience that elicits protective motor actions, results in avoidance and may modify species-specific traits of behavior, including social behavior” [5]. Since animals are unable to verbally communicate with humans, owners and veterinarians are left with the task of identifying when an animal is in pain. Signs of pain have been associated with specific behavioral indicators in the face and body in many animals from humans to livestock [7]. However, identifying pain in donkeys presents a specific challenge. Donkeys are often described as stoic animals, meaning they may endure pain or hardship without the display of feelings and complaints. Due to this behavior, a donkey may ultimately mask pain with behaviors, such as sham eating or by only showing slight indications of pain, such as a twitch to the tail or change in posture. These subtle behaviors may go unnoticed, thus concealing illness/injury [2,3,8,9]. These behavioral adaptations allow the donkey to appear normal, and likely decrease the risk of an ill/injured donkey being targeted by a predator. As useful as these behaviors and stoicism may be to a wild donkey, they make it difficult to identify pain when evaluating domesticated donkeys (e.g., working, companion or production donkeys) or feral donkeys now living in captivity.

As donkeys become increasingly popular as companion animals along with the use of donkeys for draft animals and growing use for production (milk, meat, and skin), there is a growing need to understand more about their behavior related to their overall well-being from both an owner and veterinary perspective. Previous studies have yielded facial expression ethograms [10] and composite ethograms (face and body) to assess pain in horses [5,11], as well as facial expression pain scales for donkeys and visual analogue score based on disease conditions observed [12,13]. The objectives of this study were to construct a donkey ethogram, followed by a donkey grimace scale to be applied to donkeys in order to determine whether there was a notable difference in scores pre- and post-castration, which could reveal possible discomfort/pain. The study also aimed to compare the reliability of observers with different levels of experience and knowledge with donkeys. It was hypothesized that a change would be seen in the pre- and post-castration score per donkey; more specifically, on average the post castration pictures would have higher scores, when compared to the pre-castration pictures, indicating a higher presence of pain behavior markers in donkeys post-castration. The hypothesis regarding the reliability of observers was that the more donkey experience and knowledge an observer had, the higher probability they would correctly identify pain markers in the images.

## 2. Materials and Methods

### 2.1. Animal Sample

Nine adult male standard donkeys (Standard is a breed of donkey recognized in the U.S. according to height and weight, 150 to 300 kg), ranging from 2 to 10 years old were photographed pre- and post-castration. All donkeys were clinically evaluated by the attending veterinarian (a board-certified surgeon with over thirty years of clinical experience working with and owning donkeys) and his team. Any donkey not deemed healthy based on physiological parameters and overall appearance did not undergo castration. The evaluation of pain based on the evidence reported by Dyson et al. [11], who suggested the recognition of certain behavioral features may act as potential indicators of musculoskeletal pain, which may enable the early recognition of certain distressing conditions in equids. The indicators proposed by Gleerup and Lindegaard [14] were considered as behavioral indicators of pain in the context of normal equine behavior. The donkeys were physiologically evaluated by veterinarians before their free-from-pain condition could be ensured, and these could be returned to their habitual housing. The review by Ashley et al. [2] was considered to adjust the information and protocols of identification of pain-related signs to the specific behavioral nature of donkeys. The overall appearance was evaluated following the premises in McLean et al. [15].

Donkeys were acclimated to the farm and arrived 10 days before surgery and housed in a group in paddocks with shelter, ad libitum hay, and free access to water. All donkeys were part of a private donkey rescue.

### 2.2. Photo Set and Selection Protocol

A total of 270 photographs were initially taken for the present study. All photos were taken with an Apple mini iPad (version ISO 11.4.1, Apple Computer, Cupertino, CA, USA). The photos were identified depending on whether they were taken pre- or post-castration. Five photos were taken of each donkey from the following positions: One frontal photograph, one lateral photograph from the right side, one lateral photograph from the left side, one body photograph from the right side and one body photograph from the left. Each donkey was photographed 48 h, 24 h, 0 h before surgery and 8 h, 24 h and 48 h post-surgery on the farm. Fifty-four photos were randomly selected (27 pre- and 27 post-castration) to build the materials used for the surveys as suggested by Dalla Costa et al. [16], seeking a balanced design to perform the statistical analysis.

### 2.3. Anesthesia and Surgery Protocol

Donkeys were anesthetized beginning in the morning by a standard xylazine/ketamine protocol [5]: xylazine hydrochloride 1 mg/kg and butorphanol tartrate 0.02 mg/kg IV for sedation, followed by ketamine 2 mg/kg and midazolam 0.03 mg/kg IV for induction of anesthesia. Donkeys requiring additional anesthesia to complete the procedure were given 0.5 mg/kg xylazine mixed with 1 mg/kg of ketamine. No catheter was used. All injections were administered IV in the jugular vein with a 3 cm, 18 ga needle. Flunixin meglumine (Merck Animal Health USA, De Soto, KS, USA) 1 mg/kg was administered IV intraoperatively as an analgesic drug to reduce inflammation and to relieve pain. Procaine Penicillin G (Pfizer, New York City, NY, USA) 18,000 IU/kg IM was administered intraoperatively for infection prophylaxis. All patients were placed in either right or left lateral recumbency, with the uppermost rear leg elevated and restrained with a soft rope. Their spermatic cords were identified by palpation, and 10 mL of 2% lidocaine was injected into each cord. Following a routine surgical prep using povidone-iodine surgical scrub (First Priority Inc., Elgin, IL, USA) and povidone-iodine solution rinse (First Priority Inc.), two incisions were made ventrally in the scrotum. The fascia covering the vaginal tunic was bluntly dissected, and each testis with an attached spermatic cord was exteriorized, while keeping the tunic closed (“closed technique”). The cremaster muscles were broken to decrease the diameter of the tissue to be ligated. A modification of the “Miller’s” knot was used for transfixation and ligation of the spermatic cords. The ligation process was completed by using Coated Polyglactin 910 (Coated Vicryl Plus, Ethicon, Somerville, NJ, USA) and Triclosan suture (Coated Vicryl Plus, Ethicon). The spermatic cords were crushed and transected 1 cm distal to the ligature with Serra Emasculators, and spermatic cord stumps were examined for hemorrhage. Excess loose fascia that might extend from the incision site was removed, and incision stretched to maintain postoperative drainage. All surgeries were performed on the same day with the same experienced NGO team, including donkey surgeon, team, and anesthetics, and recovered on site. The surgeries were done in the field. No complications were found, nor any of the donkeys needed redosing during the application of the castration procedures. Castration took 5 min ± 5 min to perform. The whole procedure, including preparation, surgery, and recovery, took an average of 20 min ± 5 min per donkey.

### 2.4. Procedure for Volunteer Classification, Development of Donkey Ethogram, Survey and Sample Description

#### 2.4.1. Observer Demographics

Twelve observers (41.7% males and 58.3% females) with ages ranging from 20 to 70 years volunteered to participate in the donkey ethogram training and castration image survey. Those who volunteered had various levels of donkey experience and knowledge and included veterinarians, veterinary students, researchers, and donkey owners or were a combination of the aforementioned descriptors. All levels of knowledge and experience were equally represented in their frequency in the sample. All observers were then placed into categories based on the amount of time they spent with donkeys (interaction); minimal (<4 h/month), intermediate (4–6 h/month), and extensive (>6 h/month), and knowledge: Minimal (up to 100 h of formal training, attended a symposium dedicated to donkeys or had a certified course in donkey related science), moderate (more than 100 h of formal training, attended two or more donkey symposiums or had more than one certified course in donkey related science), and extensive knowledge (more than 100 h of formal training, a degree or PhD in donkey-related science, and research/work with donkeys on an everyday basis). Observer sample distributed across knowledge and experience levels as follows—33.3% of the observers presented minimal interaction with donkeys and minimal knowledge, 25% of the observers presented intermediate interaction with donkeys and moderate knowledge, and 41.7% of the observers presented extensive interaction with donkeys and extensive knowledge. The observers in the sample were chosen seeking an almost-equal representation of sexes, interaction and knowledge levels, to prevent the bias derived from the potential overrepresentation of some categories over the rest.

#### 2.4.2. Ethogram

An ethogram displaying key body language often used to evaluate pain in other animals was developed to train observers to evaluate this body language for signs of change (Refer to Appendix A for ethogram) [10,11,15]. The training module included photographs, written descriptions, and a voice-over to describe each descriptor and photograph. Changes in tension, posture, or position were displayed in the training material and described by written and voice descriptions. Signs of discomfort or pain, such as orbital tightening of eyes, muzzle tension, along with ear and body language were illustrated and described [2,3,4,17,18,19,20] (Figure 1, Figure 2, Figure 3, Figure 4 and Figure 5). Each observer was required to complete the donkey ethogram training module before completing the study survey. From the ethogram training materials, nine facial and body language markers (ear frontal and side position, eye description (shape and tension), muzzle and nostril description (shape and tension), and stance) were then identified as possible pain indicators and were utilized for the development of the donkey grimace scale tested in the survey.

#### 2.4.3. Survey

The survey included 54 photographs taken with an Apple mini iPad, six pictures per donkey: Lateral, frontal head, and lateral body view (one pre and post castration photograph/view). Each observer took the survey once per day on their own device (e.g., computer, laptop, tablet, or smartphone), for three consecutive days to ensure intraobserver reliability. Photographs were presented in a digital form and survey to the observers in the same order on each consecutive day. Observers were blinded to the treatment of each donkey photograph, meaning the observer did not know whether each picture was pre-castration or post-castration. The database comprised of 18,072 scores.

#### 2.4.4. Sample Description

Observers were asked to score eleven parameters per photograph of each of the nine donkeys. The observer responded with a score ranging from 0 to 2 (0 being absent, 1 being moderately present, and 2 being obviously present) for each of the body language and facial parameters: Ears down, ears back, eye white showing, glazed look, orbital tightening, eyes round shape, nostril tension, eyes narrow shape, muzzle tension, and abnormal stance [18]. The observers were also given an “I don’t know” option (in addition to the 0–2 scale) which they were instructed to select if they could not identify the donkey’s response for that marker based on the image. Also, observers were asked to score their overall perception of the animal’s pain status, while being blinded to the treatment (pre or post castration) present in the images (Figure 1, Figure 2, Figure 3, Figure 4 and Figure 5).

### 2.5. Intra and Interobserver Reliability Testing

Interobserver reliability was assessed at two levels. First, the measurements of every observer were compared for days one to three, and second, measurements of scores across the total panel for the summation of all three days were compared. Intraobserver reliability was tested by comparing the responses of the same observer over three consecutive days scoring the survey. The intraclass correlation coefficient of the Reliability Analysis procedure of the Scale package in SPSS Statistics, Version 25.0, IBM Corp., Armonk, NY, USA, (2017) was used to assess inter (across observers) and intraobserver (for the observations of the same observer across the different days) reliability. Responses were then classified as successful or not, depending on whether the observers had identified signs of discomfort related to pain or not compared to pre-castration images [8,21]. Per clinical examination from attending veterinarian, all donkeys were considered to be in the absence of pain before castration, and thus, served as their control. Post castration pain status was determined after the diagnosis issued by the attending veterinarian. The pain status determined was used to evaluate the responses by each of the observers participating in the survey. The summaries of the results for inter and intraobserver reliability are shown in Appendix A.

### 2.6. Conditioning Factors Identification and Testing for the Probability of Success to Identify Post Procedure Pain-Related Signs

Before testing for the probability of success to identify signs of pain, Kolmogorov-Smirnov (Appendix A) and Levene’s test were performed to study data distribution and variance across groups to discard gross violations of normality and homoscedasticity assumptions. As both parametric assumptions had been grossly violated (*p* > 0.01), a nonparametric approach was considered. Frequency analysis and Pearson’s Chi-squared were performed to test for differences in the probability of success to identify pain signs (ability to detect true positive signs of pain, true negative signs of pain, false-positive signs of pain and false-negative signs of pain) across factors: Castration status, observer gender, donkey knowledge, donkey interaction, body language. Frequency analysis and Pearson’s Chi-squared were performed using the Crosstab task from the descriptive statistics procedure in SPSS Statistics, Version 25.0, IBM Corp. (2017). Pearson’s chi-square test is used to determine whether there is a statistically significant difference between the expected frequencies and the observed frequencies in one or more categories of a contingency table [22], with each of the observation being exclusively categorized in one category (false or true positive or false or true negative [23]). As stated in Roscoe and Byars [24], when sampling from a uniformly distributed populations (as the one in our study), the Kolmogorov test is markedly and consistently conservative even under conditions most favorable to the test, and chi-square proved to be quite robust even with very small samples. Additionally, chi-square has proven to be robust in tests of goodness-of-fit to uniform with fractional expected frequencies, as it occurs in our study.

### 2.7. Overall and Specific Body Language Sensitivity, Specificity and Accuracy for Post Procedure Pain-Related Signs Identification

Overall sensitivity, specificity, and accuracy were computed to determine the efficacy of the technique to report a comprehensive, accurate assessment of pain-related body language signs across body areas and on the whole, as a way to identify potential body language which should be considered better predictors for pain-related conditions versus those which may report misleading information. Sensitivity is the percentage of donkeys’ post castration being identified as in pain. Specificity is the percentage of donkeys pre castration being identified as not in pain. Accuracy is the likelihood of detecting either a true positive or true negative [25].

### 2.8. Declaration of Ethics

The study was reviewed and approved by the University of California Davis’ Institutional Animal Care and Use Committee, but no protocol number was given, since only observations were being conducted by the research team. Castrations were scheduled and performed by a veterinary nongovernmental organization on-farm (in the field) independent of the study. No additional permission was needed following the recommendations of Royal Decree-Law 53/2013 and its credited entity, the Ethics Committee of Animal Experimentation from the University of Córdoba, given the application of the protocols present in this study followed the premises cited in the 5th section of its 2nd article, as the animals assessed were used for credited zootechnical use.

## 3. Results

### 3.1. Intra and Interobserver Reliability Testing

All observers showed a Cronbach’s α parameter above 0.70 (average measures), suggesting that each observer reliably scored across all test days. Reliability coefficients (Cronbach’s alpha) above 0.85 are generally regarded as high and those between 0.65 and 0.85 as moderate. The total panel across all three days reported a Cronbach’s α parameter above 0.90 (average measures), suggesting total panel reliably scored across days. It was likely for the panel to score the same for the same photo on the three repeated sessions. Additionally, interobserver reliability for the whole panel was tested, and the value of 0.97 (average measures) suggests agreement between observers was excellent. Hence, any of the members is valid, reliable, and provides consistent enough measures to be considered independently.

### 3.2. Conditioning Factors Identification

There was a significant (*p* < 0.001) difference between female and male observers in their ability to correctly identify signs of pain after castration (Appendix A). The relationship between knowledge and gender was studied using partial eta squared (ηp^2^). Partial eta squared (ηp^2^) was 0.08, which meant that the effect of gender accounted for 8% of the differences in knowledge plus associated error variance. Sample size and possible factor combinations were not sufficient to justify the performance of univariate follow-up statistical analyses to determine whether significant differences could be ascribed to the interaction between knowledge and gender—however, these factors separately, as some of the categories, could be misrepresented. The study of the interaction between knowledge and gender may imply the multiplication of the number of possibilities within the interaction factor. Hence, instead of considering two variables (knowledge and gender) with two (male and female) and three levels (minimal, intermediate and extensive knowledge), respectively, we would be considering one variable (knowledge level-gender) with six levels (male-minimal knowledge, female-minimal knowledge, male-intermediate knowledge, female-intermediate knowledge, male-extensive knowledge and female-extensive knowledge). Then, in the context of our sample and to maintain the balance in the representativity of each possibility, clustering both variables under an interaction term may reduce the frequency of cases across levels to two cases per level. This offers lower chances to compare than in a case in which the variables of knowledge level and gender are considered separately [26]. Males were slightly less likely to identify a true positive (identify pain in the post castration images) and to score a false-negative (fail to identify markers considered to be pain indicators) than female observers were (Figure 6). There was also a significant difference across levels of donkey knowledge in the observers’ ability to correctly identify signs of pain in post castration images. The probability of scoring true positives was higher when observers had moderate or extensive knowledge of donkeys (Appendix A). Conversely, observers with minimal knowledge of donkeys had higher a likelihood to score false-positives or false-negatives than those with moderate or extensive knowledge of donkeys. Observers with moderate knowledge of donkeys reported a slightly higher ability to accurately detect signs of pain after castration than those having extensive knowledge and considerably higher than those having minimal knowledge. However, observers having minimal or moderate knowledge of donkeys presented a higher likelihood to make a false-negative. On the contrary, observers presenting extensive knowledge of donkeys had a higher likelihood to make false-positives (Appendix A). A significant difference across the levels of interaction with donkeys was seen in the results from the observers’ ability to correctly identify signs of pain after castration. The pattern of scoring false-negatives, true negatives/positives and false-positives between observers who occasionally (<4 h/month), sometimes (4–6 h/month), and often (>6 h/month) spent time with donkeys followed the same trend as observers with minimal, intermediate, and extensive knowledge of donkeys (Appendix A and Figure 7).

### 3.3. Testing for the Probability of Success to Identify Post Procedure Pain-Related Signs

Significant differences (*p* < 0.001) were found for the distribution of the intensity of signs identified by the observers between pre castration and post castration images (Table 1). As reported in Table 1, 2.6% and 8.7% higher frequencies were found when moderately present signs were detected, and obviously present signs were detected for post castration images, respectively. Frequencies for the absence of signs were 14.3% higher for pre castration images. ‘I don’t know’ response frequency was quite similar with only 0.30% higher frequency of ‘I don’t know’ responses from post castration pictures (Figure 8).

### 3.4. Overall and Specific Body Language Sensitivity, Specificity and Accuracy for Post Procedure Pain-Related Signs Identification

Significant differences were found for the probability of detecting signs of pain across the body language markers tested (Appendix A). A change in stance and overall appearance reported the lowest rates of false-positives and false-negatives, improving the specificity and sensitivity of the technique. Eyes presenting a narrow shape, orbital tightening, and glazed were used to correctly classify 60% of cases, however, the possibility of making a false-positive increased in comparison to false-positive and negative frequency reported for abnormal stance or overall appearance. Nostril tension and eye whites showing described the opposite trend, although they were able to correctly classify around 54% of the cases. The nostril tension marker yielded 33% of false-positive cases, while the eye white showing marker yielded 40% of false-negatives. Contrastingly, nostril tension reported the lowest rate for false-negatives (12.40%) and eye whites showing reported the lowest rate for false-positives (6.90%). The sign reporting the least accurate results was eyes round shape; only 39% were classified correctly, with 35% being false-negatives, and 25% false-positives. Our results may be in line with those for sheep by Häger et al. [27], who reported an accuracy rate of 68.2%, and a false-positive rate of 22.7% (more common than false-negatives, 9.1%). These similar results may base on the suggestions by Larrondo et al. [28], who reported an increased difficulty in pain recognition in lambs which indeed conditions the use of analgesics. A slightly higher accuracy of 73.3% was reported for a Horse Grimace Scale, which may reflect a relative higher ease to identify signs [10], which has traditionally been supported by the literature [2]. Higher values have been reported by Reijgwart et al. [29] when using their Grimace scales methods for ferrets. These authors reported sensitivity and specificity values of 85% and 74%, respectively. The highest accuracy reported (80%) was achieved when whisker retraction was excluded, and only orbital tightening was considered. Lower rates of false-negatives together with higher rates of false-positives may be indicative of a rather cautious assessment of pain, due to the difficulty to identify pain-related behavioral signs in donkeys. Our results may complement those reported by Dai et al. [30], as it may be the ability of panel training methods to inform about the subtle signs reported by animals rather than the time needed to train the members of the panel, especially for signs, such as tension above the eye area, prominent strained chewing muscles, mouth strained and pronounced chin and strained nostrils. This could also be supported on the higher accuracy found in our DGS (Donkey Grimace Scale) for overall appearance and abnormal stance, which may indeed reflect a higher interobserver agreement, and indirectly indicative of a successful training period. Body language markers that have sensitivity or specificity below 50% should not be considered when aiming at assessing pain (for instance, eyes round shape) (Figure 9). The sensitivity of post castration images was 63.18% with detected pain. Significant differences were the distribution of the intensity of signs identified by observers between pre-castration and post-castration images. Higher frequencies of pain markers were found when moderately (2.6%) and obviously present signs (8.7%) were detected for post castration images. Table 2 reports a summary of the results for sensitivity, specificity and accuracy of each body language as a pain indicator and of the overall donkey grimace scale.

## 4. Discussion

The original lack of scientific support in regard to the assessment of specific donkey welfare contrasts the advances that have recently appeared in the scene. This methodological revolution appeared as an effort to address the misattributions and misconceptions of the donkey species. Incorrect evaluation of donkey welfare often bases on an incorrect perception of the behavioural, physiological, pathological idiosyncrasies of the species, among others. In this context, complex validated welfare approaches [31] can benefit from the methodologies implemented by large scale applicable methods [32] to report replicable and still trustable results in contexts, such as medium- or low-income countries, for which donkey is a common and relatively frequent element. The present research can complement the aforementioned methods as, it focuses on the human ability to determine signs of pain, which according to literature has often been deemed difficult provided to the subtleness of donkeys to display signs of distress [2].

Castration is considered to be one of the most common surgical procedures practiced by veterinarians [10]. The present study measured indicators of pain in donkeys before and after the procedure by observing changes in facial units and body language with the objective that this tool could be easily applied and used to identify pain in donkeys. Previous studies have sought to identify indicators of pain in equids. One study measured similar facial action units in horses before and after castration and created a grimace scale to assist with recognizing signs of discomfort and pain, but this has not been achieved in donkeys until this study was conducted [10]. The horse study assessed pain in similar areas to this study, such as orbital tightening, muzzle and ear position and similar indicators were observed in donkeys in this study in the same facial areas.

Ashley et al. [2] reported the present paucity of scientific research into equine pain behavior in spite of the relevance that it is attributed to it in horse clinics. In their review involving papers published during a period of 20 years, the same authors would state, this lack of scientific attention to the horse’s pain-related behavior is extensive and even less common in regards other equine species, such as the donkey or its hybrids. This becomes explicit in the review as among the nonspecific behavioral indicators of pain in equids, the review by Ashley et al. [2] offers an extensive description of horse pain-related behavior, such as considerable restlessness, agitation and anxiety, rigid stance and reluctance to move, lowered head carriage, fixed stare and dilated nostrils, clenched jaw, aggression towards own foal, handlers, other horses, objects and self. Oppositely, the same authors claim these signs cannot either be relied upon, are often misdiagnosed, are difficult to differentiate, are not reported or described, or appear in a more subtle display for the donkey, respectively. In regard to the signs of abdominal pain, the review reports deep groaning or vocalization, rolling, abdominal kicking, flank watching, stretching and dullness and depression, in horses. Oppositely, the first four aforementioned behavior was not reported or have not been described in the donkey, while, stretching, which has importantly been related to other signs with the purpose to evaluate analgesia, has been nonspecifically been ascribed to colic, and is often accompanied by difficulty in defecating or urinating. Dullness and depressive-like-behaviors have been commonly reported as the only observable pain-related behaviors in donkeys, which is often simultaneous to lethargy and reduced alertness, self-isolation or handlers’ contact reluctance.

Among the behavioral indicators of limb and foot pain in equids collected by Ashley et al. [2] in their review mention weight-shifting between limbs, limb guarding, abnormal weight distribution, pointing, hanging and rotating limbs, abnormal movement and reluctance to move in horses. Later, the authors add these patterns involving the limbs and foot are normally unreported or not commonly seen in donkeys (we understand, if they are experiencing the same level of pain). Furthermore, these authors suggest literature may have reported reluctance to move to be weakly associated with limb pain in the donkey, while repeated episodes of lying down may offer a rather evident sign of the donkey being in distress. Last, Ashley et al. [2] report the set of most frequent behavioral indicators of head and dental pain in horses and donkeys may involve headshaking, abnormal bit behavior, altered eating; anorexia, quidding, food pocketing. Particularly, quidding partially chewed portions of feed may be indicative of severe chewing discomfort and can particularly lead to choking in older donkeys. Contextually, the information reported shows the noticeable lack of specific or updated information in regards how pain is behaviorally expressed by donkeys, which makes the development of a donkey grimace scale necessary in order to avoid the misconception of pain-related signs as a result of an incorrect comparison with other affine species, such as horses.

Another study measured pain thresholds in donkeys and horses with mechanical stimuli applied to their limb and recorded behavioral responses; the animals served as their own controls [3]. In the present study, a donkey grimace scale was developed and applied to pre- and post-castration images to measure a difference in facial action units and body language. A recent study included a facial grimace scale for donkeys experiencing acute and chronic pain from various anatomical locations (such as the head, limbs, and abdomen), and, compared their signs of pain with donkeys, assumed to not be in pain [1]. Granted, pain behavior may be dependent upon individual animal basis, and signs of discomfort may vary depending on the type of pain if it is acute versus chronic. Accordingly, donkeys in the present study served as their own control, following the model used in some previous pain studies [3,11,33]. When using the same donkey as a self-control, facial expression and body language could be observed and recorded before a painful procedure, castration, and after; thus, emphasizing changes in behaviors before and after the procedure. In donkeys, signs of discomfort and pain may be subtler; therefore, one must consider the whole donkey to be able to accurately identify the presence of pain [1,3,4]. Furthermore, being able to recognize these subtle differences may prove to be challenging to those who have limited experience working with donkeys. Regan et al. [3] observed that painful donkeys would swish their tails, hold their ears to the side (facing down) and exhibit changes to head posture. The ethogram created from the present study suggested that observing the overall body posture and reactions of donkeys may improve our chances of recognizing pain in donkeys, as has been indicated in a study by Grint et al. [4]. The results of the present study indicated there was a statistically significant difference in observer scores when using the donkey grimace scale for both the face and body between castration images. All observers had undergone an extensive training process to recognize signs of donkey discomfort from observations in facial action units and body stance and/or posture, thus increasing the likelihood that observers would be able to identify subtle pain indicators. Observer training is strongly recommended to assess and improve interobserver reliability if necessary and possible before trials start [26]. As suggested by Bell et al. [27], an improvement of interobserver reliability (ICC, κ or Cronbach’s α) from 0.50 to 0.80 (below the levels found in the present study) within 3 to 5 training sessions (three sessions were used in our study) may be associated with a reduction of approximately 40% of the number of patients in each treatment group required to show a particular true score group difference. In a recently published study, researchers were able to correlate a change in donkeys’ facial grimace, behavior, and physiology with head pain, lameness and post colic surgery [1,3,4]. A previous study observed donkeys during castration surgery as having the same or greater amount of pain when compared to ponies [5]. Other studies have indicated that many mammals, including sheep, pigs, cows, and horses, experience some degree of pain after castration [17,18,19,20]. Therefore, it was assumed that post-castration donkeys were not completely comfortable, despite receiving an analgesic drug, and that changes in scores from the donkey grimace scale seen after castration could be indicators of pain. One of the most accurate body language markers we found for detecting distress related to pain was the overall appearance of the donkey and body stance. Our study’s results indicate the importance of observing more than just the face of the donkey to identify signs of pain. Facial indicators, such as ears, eyes, nostrils may be more easily influenced by external factors, such as other donkeys and the environment of the donkeys, thus it is important to look at the donkey as a whole for signs of behavior related to pain [21]. In previous studies, indicators of a donkey in pain or sick may include the following behavioral signs associated with pain, such as a slight twitch to the tail, but authors claimed they were subtler than those of horses [2,9,10,34,35,36]. Several studies have confirmed that indicators of a donkey in pain may include sham eating, chewing, generalized dullness, shifting weight to contralateral limb, unresponsive/decreased mobility of ears, twitching tail, and tail tucked [3,4,5]. The present study confirmed the importance of observing the overall appearance of the donkey to accurately identify pain. However, since the present study examined pictures and not the video of the donkeys, it was difficult to monitor ear mobility or sham eating.

Some studies have developed and tested a Horse Grimace Scale [10], an ethogram of facial expressions of ridden horses [36] and applied this ethogram to differentiate between sound and lame ridden horses [37]. Another study developed an ethogram for the body, face, and gait of the horse and aimed to differentiate between lame and non-lame ridden horses [11]. Similarly, to the aforementioned study, the current study also aimed to distinguish pain indicators based on the whole body in addition to facial indicators. Contrastingly, the present study examined donkeys with post-operative pain instead of horses with musculoskeletal pain. Although multiple studies have utilized facial action units to recognize signs of pain in horses, therefore, suggesting a similar scale, such as the scale used in this study could be effectively applied and used in donkeys. Recent studies have applied various pain scales, such as a Facial Expression Pain Scale, EQUUS-DONKEY-COMPASS and EQUUS-DONKEY-FAP, to groups of donkeys experiencing different kinds of pain, including acute orthopedic pain, colic, head-related pain, and post-operative pain and applied the same scales to a control group of healthy donkeys. Although these studies did not yield significant results in regards to post-operative donkeys, likely due to the variation in how long the donkey may have been experiencing pain or the stage of the disease, both studies yielded significant results for facial pain, colic, and lameness [6,12], and provided a basis for the implementation of a grimace scale to be used to identify pain indicators in donkeys.

Donkeys in the current study served as self-controls. The decision for using donkeys as self-controls was supported in previous studies for three main reasons. First, for ethical reasons, as well as to adhere to a high standard of welfare, a control group of donkeys undergoing castration without analgesic treatment was not used in this study [10]. This is based upon the claims made on Mair et al. [38], who reported that providing no analgesia for castration in equines is inhumane.

The second main reason this study was designed using donkeys as their own self-control was due to findings in previous donkey studies attempting to measure and define pain behavior and measure the clearance of analgesic agents. Donkeys were used as self-controls in a study that tested for the palliative effect of detomidine as a sedative and analgesic in cases where severe pain was reported during abdominal colic [39]. The severity of abdominal pain accompanying colic was determined prior to (time 0) and after drug administration (15, 30, 45 and 60 min) using a standardized scoring system to describe several clinical parameters common to equine colic. Among these clinical parameters, the presence or absence of body signs was considered using a scale from no evidence of signs of discomfort to severe evidence of signs of discomfort. This study, similar to our study, photographed the donkeys as various times [40]. For our study, additional times were set to take photos prior to castration, as a way to monitor the animals to make sure no evident changes occurred prior to castration at 48 h, 24 h, 0 h before surgery. Amin and Najim [40] also compared detomidine, ketamine and other anesthetics stated that full recovery was achieved at 40 min in donkeys. In regard to the metabolism of analgesics, donkeys may describe two different trends, one characterizing the period of time that takes for half-life to be reached, then afterwards the period of time needed for the analgesic to disappear from the body or at least its effects to not be present. Contextually, Grosenbaugh et al. [41], reported NSAIDs may be clinically effective longer in donkeys than in horses as indicated by the respective plasma t1/2 (half time in donkeys ranges from 0.75 to 4.5 h, with maximum limit only being half an hour longer than in horses). The study suggested a faster reduction of drug plasma levels after the drug had reached half its concentration (half-life) provided the notably increased clearance times of 1.78 (ml/kg bwt/h), which almost double those in horses [2]. Bearing this in mind and to ensure the effects of any of the drugs did not alter the donkeys’ behavior, post castration photos were taken 8 h, 24 h and 48 h post-surgery. Another comparison to this study was the differences in scales—our study developed a scale that included a “don’t know” possibility to be able to gather those cases in which observers were not able to identify the degree of expression of a certain body sign related to pain which was not included in other studies [41].

Finally, we considered a previous study [10], which indicated that horses receiving varying levels of flunixin meglumine as analgesic post castration still had a significantly higher composite pain scale score and horse grimace scale score than the respective scores of the control group that did not undergo a surgical procedure; furthermore, there was no significant difference in horse grimace scale score or composite pain scale score between the two treatment groups that received different amounts of flunixin as an analgesic at varying time intervals. Another option that may have been considered for a control would have been to have one group of donkeys undergo analgesic, but no surgery procedure and another group complete the surgery. However, based on the fact other studies had used a similar protocol to this study (animals serving as self-controls pre-/post-surgery) we did not see it necessary nor in the best interest of the donkey’s welfare to place them under general anesthesia to simply photograph their face pre and post sedation.

Behavior and images for each donkey were recorded before the procedure and then after. Each donkey underwent the same medical procedure at the same time so the differences in scores and subsequently pain indicators should be due, in the majority, to the individual donkey’s acute response, thus effects derived from factors other than the procedure itself were minimized. Often donkeys will not exhibit signs of pain or discomfort until the disease and/or condition is in advanced stages or is a chronic condition (e.g., laminitis), therefore the subtle signs of pain may go unnoticed. Even when the disease or condition becomes chronic, the signs may remain unnoticed, due to the donkey having become habituated to the discomfort. This study began to define signs of discomfort related to pain in an acute phase versus chronic, in hopes of providing a tool to assist practitioners and owners in noticing these subtle behavioral changes and ultimately lead to providing treatment for pain [2,3,4,9,10,34,35,36].

The ability to recognize signs of pain in this study was significantly dependent on gender (Appendix A), experience or interaction with donkeys (Appendix A), and knowledge level of observers (Appendix A). These factors impacted the observer’s ability to correctly score a post castration donkey as painful. Previous studies have indicated that female respondents, when surveyed, tend to provide a significantly higher pain score among cattle, dogs, cats, and horses across a variety of conditions and procedures, however, the studies in which this was observed did not include whether the scores of male versus female participants were more accurate [42,43,44]. The present study found a significant difference between the ability of male versus female observers to identify pain indicators after castration. More specifically, female observers were more likely than male observers to provide a higher score for post castration pictures. Furthermore, female observers were less likely to score a false-positive (identify pain markers in pre castration pictures—control group).

Results of the present study indicated that observers with more donkey experience and knowledge are less likely to miss indicators of pain (score false-negatives). This is likely because by spending more time observing donkeys, one may become more skilled at noticing behavioral deviations from “normal”, and therefore, be more likely to pick up on subtle signs. These results suggest that practitioners and owners can become more successful at identifying when a donkey is painful by spending more time observing and becoming familiar with donkey behavior in general. Previous studies examining donkey behavior have confirmed that because donkeys demonstrate different clinical signs of pain/illness when compared to horses, previous knowledge and an understanding of the donkey can be crucial to correct examination, diagnosis and medical treatment of the donkey [9,10,21,35,36]. Another study indicated that painful conditions in donkeys, such as laminitis, are often overlooked by the owner and veterinarian, due to the donkey’s stoicism [44]. One survey of donkey owners and veterinarian surgeons investigated how pain is identified and then treated in donkeys [33]. The study found that a majority of the responses from veterinarians administered horse rate and doses of analgesic drugs to control pain in donkeys which likely is not effective. Therefore, in addition to considering the variations in the obviousness of pain behavior between horses and donkeys, once a veterinarian can identify clinical signs of a painful donkey, the treatment of whatever ails the donkey must likely be approached slightly differently to the way it would be treated for a horse [9,10,21,35]. When pain is identified in donkeys, it is possible that therapeutic levels of analgesic drugs are not met, due to species differences; previous studies have shown that donkeys have the ability to metabolize analgesics at a faster rate than a horse of similar size [45,46].

Practitioners and owners should be aware of differences between donkey and horse medication dosages, and appropriate donkey dosages should be administered [9,10,21,35,36]. The present study provided all donkeys with an injectable nonsteroidal anti-inflammatory and analgesic drug, flunixin meglumine, during the surgery. This may have influenced pain indicators seen post-surgery, though donkeys clear the drug much more rapidly than horses or mules. The sedative, xylazine, the sedative analgesic drug, butorphanol, and the dissociative anesthetic, ketamine, also have short term analgesic effects [47]. While these drugs might affect the donkey’s pain perception, they would be unlikely to change responses 24 h after surgery [47]. For this reason, even with the administration of this commonly used analgesic drug to the donkeys in this study, signs of discomfort and pain were still noted as early as eight hours post-castration. It is important to reiterate that the donkey’s stoicism does not diminish the donkey’s pain or distress, hence it is important for owners and veterinarians to observe what normal donkey behavior looks like. The present study has shown that there are behavioral indicators of pain in faces and bodies of donkeys (abnormal stance and overall appearance of the donkey proved to be particularly accurate); identification of these markers may require observation by a more experienced observer familiar with donkeys.

## 5. Conclusions

The ability to recognize subtle body language signals may prove to be challenging to those who have limited experience working with donkeys. Observing the overall body posture and reactions of donkeys may improve our chances of recognizing pain in donkeys. This indicates the importance of observing more than just the face of the donkey to identify signs of pain. Facial indicators, such as ears, eyes, and nostrils, may be more easily influenced by external factors, such as other donkeys and the environment of the donkeys, thus it is important to look at the donkey as a whole for signs of behavior related to pain. Gender, experience or interaction with donkeys, and knowledge level of observers impacted the observer’s ability to correctly score a post castration donkey as painful. Female observers were more likely than male observers to provide a higher score for post castration pictures and were less likely to score a false-positive. By spending more time observing donkeys, one may become more skilled at noticing behavioral deviations from “normal”, and therefore, be more likely to pick up on subtle signs. Conclusively, identifying the appearance of a normal donkey may assist in decreasing the number of donkeys that go untreated, due to the subtleness of their behavioral pain indicators, as the biggest obstacle of treating pain in donkeys may be identifying the pain in the first place.

## Figures and Tables

**Figure 1 animals-10-01411-f001:**
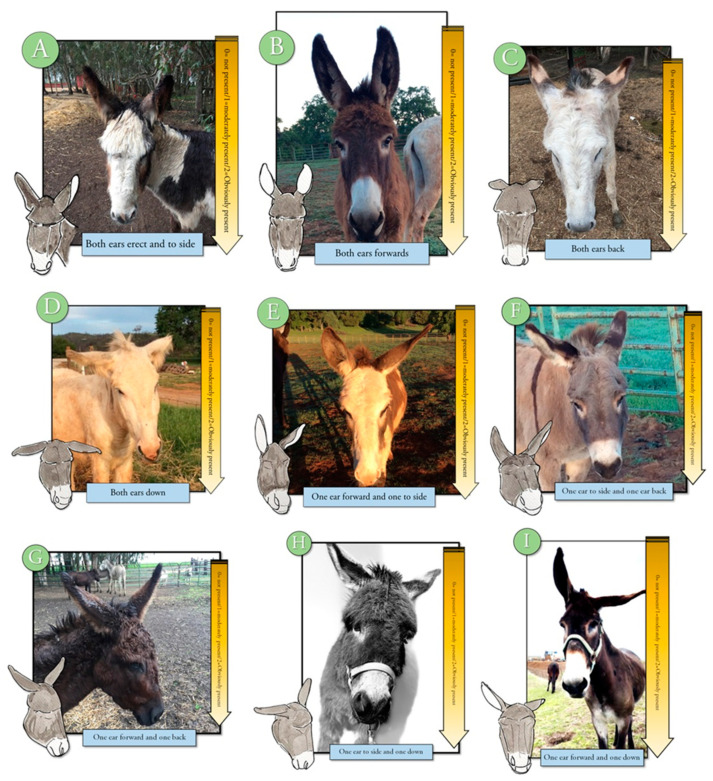
An ethogram describing ear position related to grimace scale (0—not present, 1—moderately present, and 2—obviously present). Ear positions that maybe associated with pain or discomfort include both ears back (**C**), down (**D**), one ear forward, one to the side I, One ear to the side and one back (**F**), one ear to the side and one down (**H**), and one forward and one down (**I**) along with other facial action units associated with pain. Ear positions that are less likely to be associated with pain would include both ears are erect to side (**A**), both ears are forward (**B**), one ear forward and one to the side (**E**), and possibly one ear forward and one ear back (**G**).

**Figure 2 animals-10-01411-f002:**
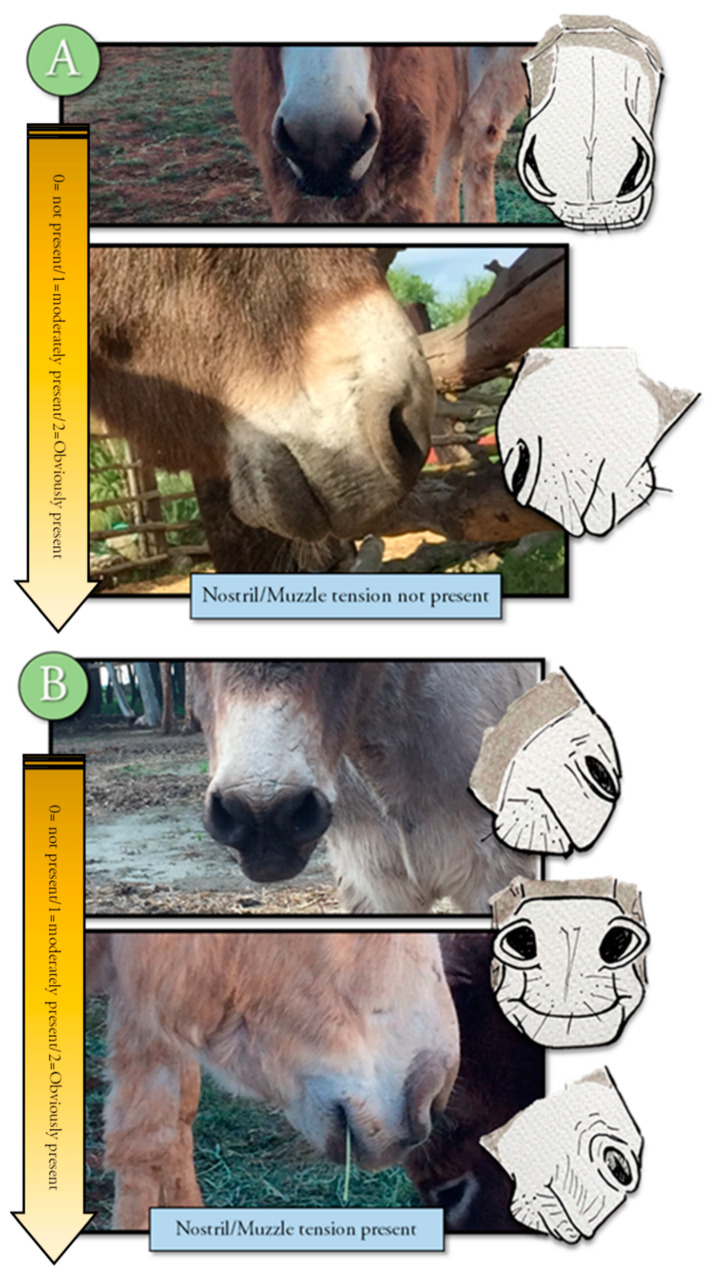
Ethogram showing nostril and muzzle tension presences. Tension is not present in picture (**A)** meaning the donkey is likely not in pain or discomfort. The tension associated with pain can be found in picture (**B**).

**Figure 3 animals-10-01411-f003:**
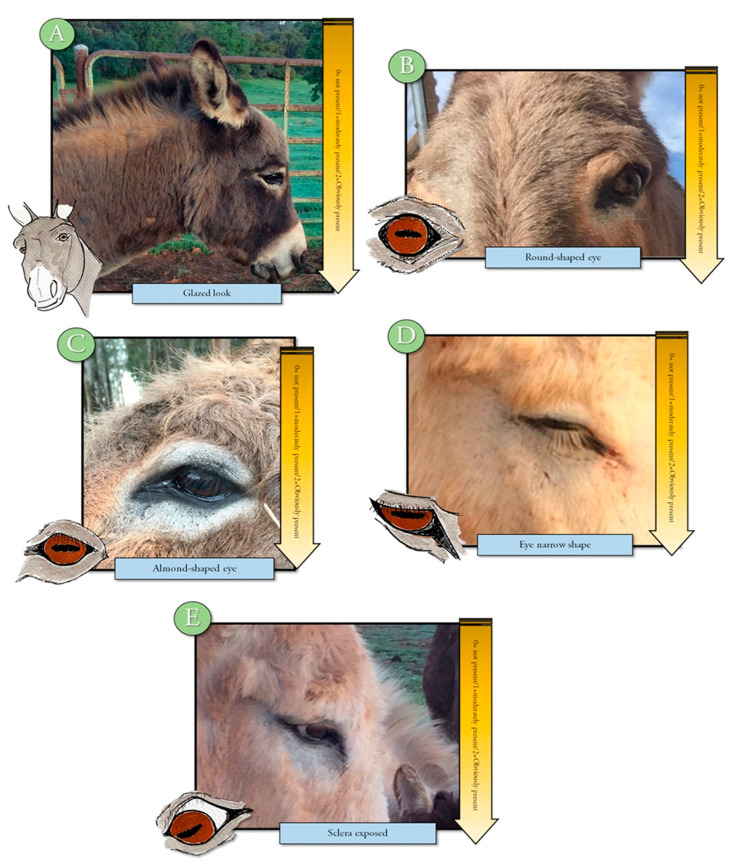
Ethogram displaying eye shape related to pain (0—not present, 1—moderately present, and 2—obviously present) with (**A**), (**D**), and (**E**) suggesting discomfort or pain. Eyes that are round in shape (**B**) and almond shape (**C**) are likely not in pain or discomfort.

**Figure 4 animals-10-01411-f004:**
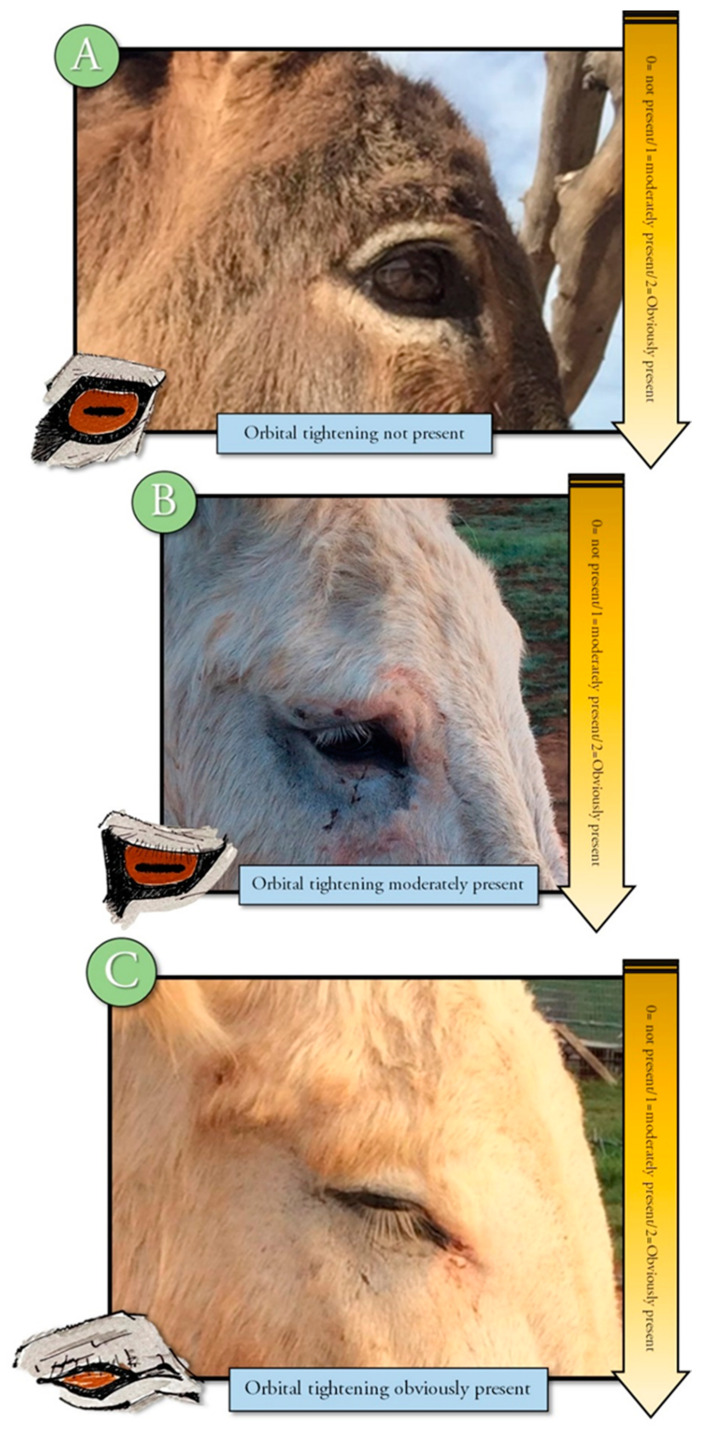
Ethogram showing orbital tightening present not present ((**A**), pain grimace score 0), moderately present ((**B**), pain score 1) and tightening present ((**C**), pain score 2).

**Figure 5 animals-10-01411-f005:**
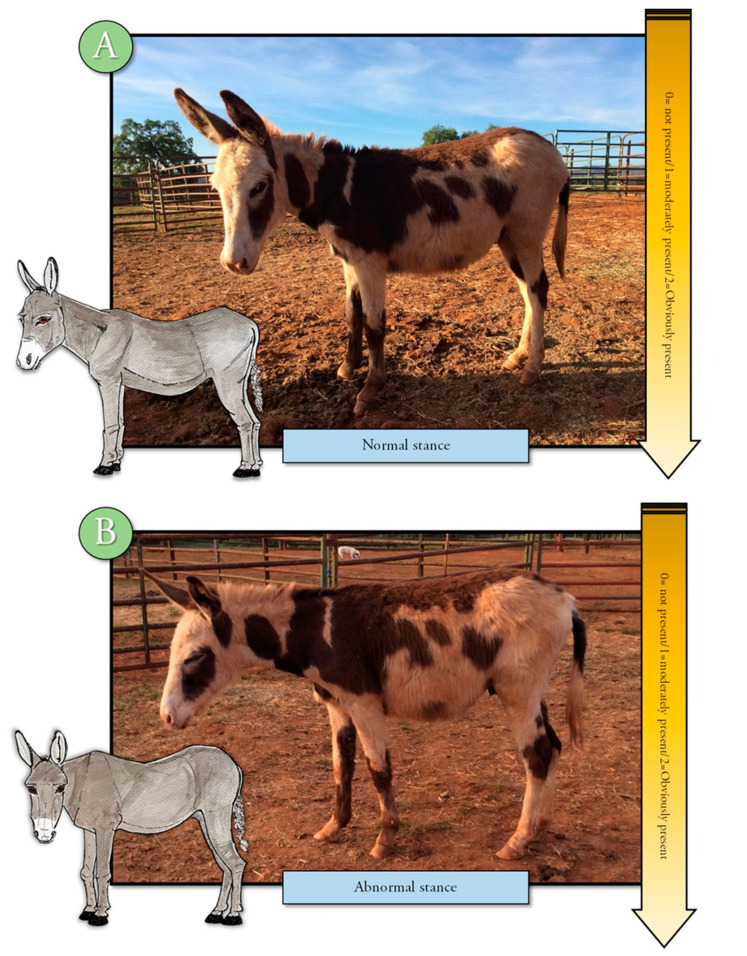
Ethogram displaying body stance, normal (**A**) and abnormal (**B**) which may be associated with discomfort or pain.

**Figure 6 animals-10-01411-f006:**
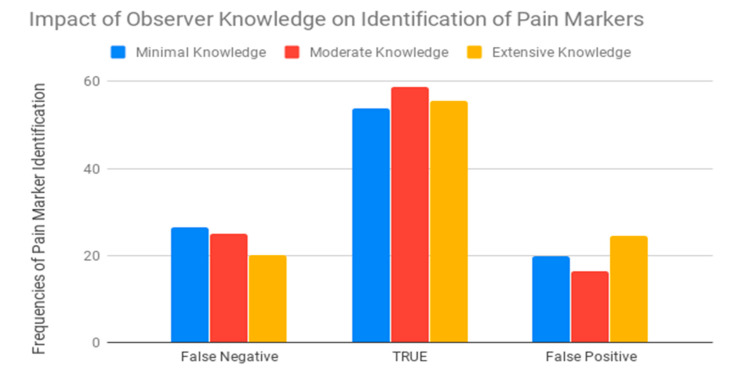
Frequencies of pain identification success possibilities across the different levels of donkey knowledge from minimal, moderate to extensive in both pre-and post-castration images. In medical diagnosis, test sensitivity represents the ability of a test to correctly identify those with the disease (true positive rate), whereas test specificity is the ability of the test to correctly identify those without the disease (true negative rate). Hence, the bar representing true values provides an overall direct measure of the quality (sensitivity/specificity) of the test in comparison to its potential drawbacks (false-positives or -negatives).

**Figure 7 animals-10-01411-f007:**
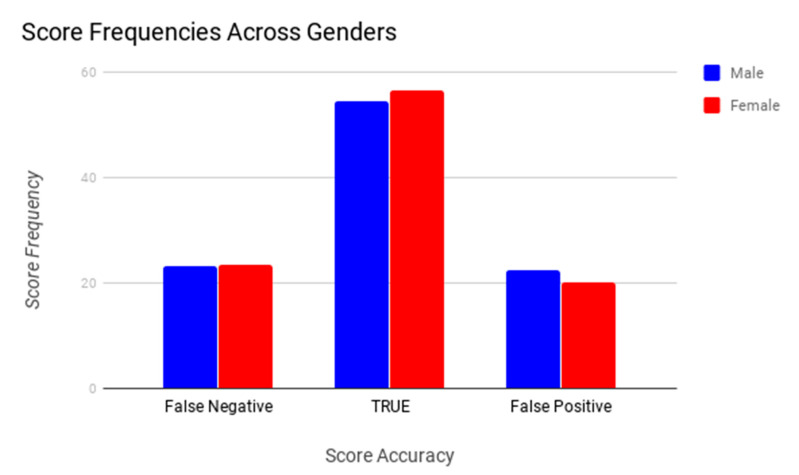
Frequencies of pain identification success possibilities across genders (male and female observers) identifying false-negative signs of pain, actual/true signs of pain and false-positive signs of pain (*p*- value 0.001). In medical diagnosis, test sensitivity is the ability of a test to correctly identify those with the disease (true positive rate), whereas test specificity is the ability of the test to correctly identify those without the disease (true negative rate). Hence, the bar representing true values provides an overall direct measure of the quality (sensitivity/specificity) of the text in comparison to its potential drawbacks (false-positives or -negatives).

**Figure 8 animals-10-01411-f008:**
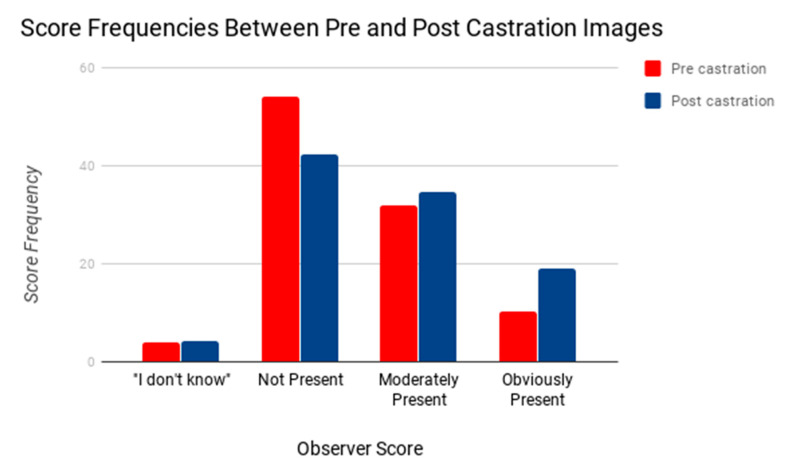
Comparing score responses related to signs of pain identification in pre- and post-castration images.

**Figure 9 animals-10-01411-f009:**
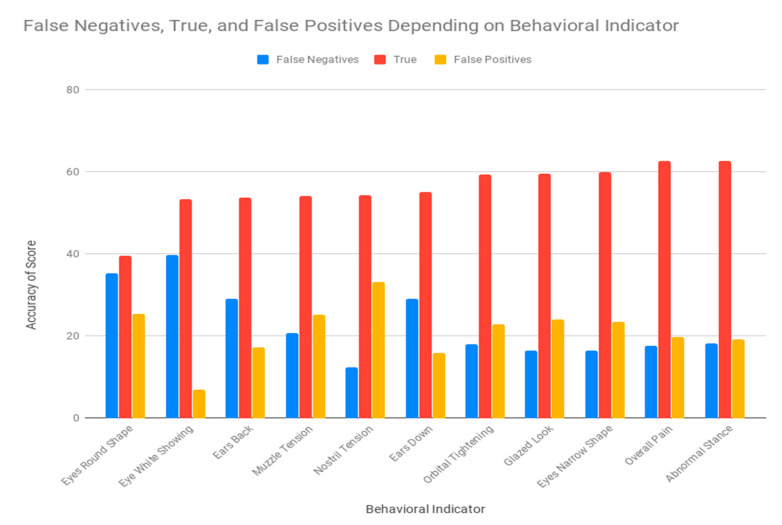
Accuracy comparison of frequency scores (false-negative, true, or false-positives) for different facial and body parameters related to signs of pain. Pre- and post-castration results were considered as the figure represent the quality of the donkey grimace scale used. In medical diagnosis, test sensitivity is the ability of a test to correctly identify those with the disease (true positive rate), whereas test specificity is the ability of the test to correctly identify those without the disease (true negative rate). Hence, the bar representing true values provides an overall direct measure of the quality (sensitivity/specificity) of the text in comparison to its potential drawbacks (false-positives or -negatives).

**Table 1 animals-10-01411-t001:** Summary for the results of frequency analyses and frequency differences between pre and post castration conditions.

Score Frequencies	I Don’t Know	Not Present	Moderately Present	Obviously Present
Pre-castration	3.90%	54.00%	32.00%	10.20%
Post-castration	4.20%	42.30%	34.60%	18.90%
Frequency differences	Value	df	Asymptotic *p*-value (2-sided)	
Pearson Chi-Square	373.496	3	0.001	

**Table 2 animals-10-01411-t002:** Summary for the results of sensitivity, specificity and accuracy for each body language sign and overall donkey grimace scale.

	Sensitivity	Specificity	Accuracy
Ears Down	54.04%	56.91%	55.10%
Ears Back	51.51%	56.86%	53.65%
Eye White Showing	52.05%	59.88%	53.40%
Glazed Look	61.18%	58.26%	59.50%
Orbital Tightening	61.17%	57.70%	59.30%
Eyes Round Shape	41.24%	36.91%	39.50%
Nostril Tension	57.68%	53.18%	54.50%
Eyes Narrow Shape	61.77%	58.84%	60.10%
Muzzle Tension	54.51%	53.76%	54.10%
Abnormal Stance	60.91%	64.25%	62.70%
Overall Appearance/Donkey Grimace Scale	63.18%	62.07%	62.60%

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
