# Peer review of "Development of a Donkey Grimace Scale to Recognize Pain in Donkeys (Equus asinus) Post Castration"

_animals, 2020, doi:10.3390/ani10081411_

Round 1

Reviewer 1 Report

This paper addresses an important question, namely, the need to develop a grimace scale for donkeys that could be used to evaluate pain responses.  Developing such a scale is certainly valuable, and the paper should be of interest to readers of the journal. 

It does, however, need some revision. 

My first general criticism is that although the paper states several times that donkeys are different from horses in their responses to pain (appearing to be more stoic, for example), there is not enough discussion of what we do know about horses or donkeys. It would help if the authors explained more clearly what is known about horse responses to pain, and then compared this to donkeys. Moreover, despite saying that they are different, I was struck by similarities - shape of the nostril, for example. 

Secondly, the numbers of both donkeys and observers was very low. As such, it makes data analysis and evaluation difficult.  It is not helpful to express percentages of observers, when there are only 12!  Why not simply say how many people?  Given the low N's,  I was doubtful about some of the statistics and conclusions.  Most notably, the claim that there are significant gender differences by observers seems far-fetched, given what is presented in the table and the diagram. The histogram does not appear to indicate much difference at all.

Thirdly, the captions to the photographs need editing.  Figures 1-3 do not include the rating scale, although that is mentioned in the caption.

Fourthly, the paper would benefit from more discussion of the implications and applications of developing a pain scale specific to donkeys

More specific points:

There are several mistakes in the English. The most pertinent is that 'painful' is used incorrectly. A donkey might be 'in pain'; it would be the stimulus that is painful.

Line 73: makes reference to growing number of donkeys kept for leisure purposes.  What about the large number of working donkeys in the world?

Line 234: Diagnosis, not diagnose

Line 294: This is not clear

Line 358: there seems to be a relatively high number of false negatives and positives;  this needs more discussion if readers are to be convinced of the claims made

Line 415: should be 'were' not 'was'

Line 451: discussion of ethics and experimental controls. In principle, a control group receiving anesthesia without surgical intervention is possible. Please add a comment.

Author Response

Reviewer #1

This paper addresses an important question, namely, the need to develop a grimace scale for donkeys that could be used to evaluate pain responses.  Developing such a scale is certainly valuable, and the paper should be of interest to readers of the journal. 

It does, however, need some revision. 

My first general criticism is that although the paper states several times that donkeys are different from horses in their responses to pain (appearing to be more stoic, for example), there is not enough discussion of what we do know about horses or donkeys. It would help if the authors explained more clearly what is known about horse responses to pain, and then compared this to donkeys. Moreover, despite saying that they are different, I was struck by similarities - shape of the nostril, for example. 

Response: Thank you for your suggestions and we have tried to include behaviors that have been documented in donkeys (e.g. sham eating, tail tuck, ears facing down) as well as some anatomical features that are different in donkeys. Additionally, we added a set of paragraphs including one of the most complete pain related behavioral sign comparisons between horses and donkeys until the date (Ashley et al, 2005). Ideally, a study comparing horses or ponies and donkeys by measuring their responses could better address the similarities such as the study conducted by Grint et al., 2017. Unfortunately, to compare similarities of anatomy such as the orbital area where we recognize the orbital socket and bone structure is much thicker and denser in donkeys compared to horses, may increase the difficulty to notice tightening. Also, the nostrils in donkeys compared to horses are usually more oval and smaller in diameter and we know this because in general we use a smaller nasal gastric tube in donkeys compared to horses. Granted, their ears are larger and possibly more expressive but that’s part of the reason we conducted this study to compare ear position and posture to discomfort. Several references are made in general about donkeys’ possible ability to hide pain as a part of a survival technique associated with the fight mechanism. Again, this is a great suggestion and we have tried to compare what we know about donkey behavior associated with discomfort and what we have learned in this study to help both owners and veterinarians identify pain in donkeys sooner than later.

Secondly, the numbers of both donkeys and observers was very low. As such, it makes data analysis and evaluation difficult.  It is not helpful to express percentages of observers, when there are only 12!  Why not simply say how many people?  Given the low N's,  I was doubtful about some of the statistics and conclusions.  Most notably, the claim that there are significant gender differences by observers seems far-fetched, given what is presented in the table and the diagram. The histogram does not appear to indicate much difference at all.

Response: In respect the number of observers, as already reported in the paper, all observers showed a Cronbach’s α parameter above 0.70 (average measures), suggesting that each observer reliably scored across all test days. Reliability coefficients (Cronbach's alpha) above 0.85 are generally regarded as high and those between 0.65 and 0.85 as moderate. The total panel across all 3 days reported a Cronbach’s α parameter above 0.90 (average measures), suggesting total panel reliably scored across days. It was likely for the panel to score the same for the same photo on the three repeated sessions. Additionally, interobserver reliability for the whole panel was tested and the value of 0.97 (average measures) suggests agreement between observers was excellent. Hence, any of the members is valid, reliable, and provides consistent enough measures to be considered independently.

This means just a single observation by a single observer (any of the ones used in the study) on any of the days during which the evaluation was carried is enough to obtain the same result on average. Hence, the conclusions obtained are supported on a solid statistical basis. The panel was trained. As it was also stated in the paper, all observers had undergone an extensive training process to recognize signs of donkey discomfort from observations in facial action units and body stance and/or posture, thus increasing the likelihood that observers would be able to identify subtle pain indicators. If scale internal consistency is ensured, observer training enables sample size reduction, for instance we add similar studies with a similar number of trained observers.

  1. Huizing, A.R.; Hamers, J.P.H.; Gulpers, M.J.M.; Berger, M.P.F. A Cluster-Randomized Trial of an Educational Intervention to Reduce the Use of Physical Restraints with Psychogeriatric Nursing Home Residents. Journal of the American Geriatrics Society 2009, 57, 1139-1148, doi:10.1111/j.1532-5415.2009.02309.x.

Sample of trained observers of 11.

We decided not to include this reference as we fell it is not necessary to justify the used of our sample.

Observer training is strongly recommended to assess and improve interobserver reliability if necessary and possible before trials start [26]. As suggested by Bell et al. [27], an improvement of interobserver reliability (ICC, κ or Cronbach’s α) from 0.50 to 0.80 (below the levels found in the present study) within 3 to 5 training sessions (three sessions were used in our study) may be associated with a reduction of approximately 40% of the number of patients in each treatment group required to show a particular true score group difference.

  1. Huizing, A.R.; Hamers, J.P.H.; Gulpers, M.J.M.; Berger, M.P.F. A Cluster-Randomized Trial of an Educational Intervention to Reduce the Use of Physical Restraints with Psychogeriatric Nursing Home Residents. Journal of the American Geriatrics Society 2009, 57, 1139-1148, doi:10.1111/j.1532-5415.2009.02309.x.
  2. Bell, M.; Milstein, R.; Beam-Goulet, J.; Lysaker, P.; Cicchetti, D. The Positive and Negative Syndrome Scale and the Brief Psychiatric Rating Scale. Reliability, comparability, and predictive validity. The Journal of nervous and mental disease 1992, 180, 723-728, doi:10.1097/00005053-199211000-00007.

We added these two reference to justify the use of our observer and animal samples and their adequacy.

Thirdly, the captions to the photographs need editing.  Figures 1-3 do not include the rating scale, although that is mentioned in the caption.

Response: Thank you for your comments. The rating scale can be found in Figure 1 on the right side in yellow, Figure 2 on the left side in yellow and in Figure 3 on the right side in yellow. Captions have been updated to include greater description in relationship to the facial and overall body expression in relationship to pain and discomfort.

Fourthly, the paper would benefit from more discussion of the implications and applications of developing a pain scale specific to donkeys

Response: Thank you for your comment and suggestion. We have tried to include the importance of developing the ethogram to help both owners and veterinarians identify pain in donkeys especially in Lines 431 to 434 and 605 to 609. “The present study has shown that there are behavioral indicators of pain in faces and bodies of donkeys (abnormal stance and overall appearance of the donkey proved to be particularly accurate); identification of these markers may require observation by a more experienced observer familiar with donkeys.” Along with the conclusion, lines 614 to 619. Also, in lines 561-563, we discuss the ability to identify acute vs chronic pain as well as how this tool can “assist practitioners and owners in noticing these subtle differences. Last but not least additional text has been added to the beginning of the discussion to emphasis the importance of this tool for application in identifying pain in donkeys, lines 393-395 “with the objective this tool could be easily applied and used to identify pain in donkeys.”

More specific points:

There are several mistakes in the English. The most pertinent is that 'painful' is used incorrectly. A donkey might be 'in pain'; it would be the stimulus that is painful.

Response: Thank you for your suggestion. We have reviewed the manuscript and made changes from painful to pain.

Line 73: makes reference to growing number of donkeys kept for leisure purposes.  What about the large number of working donkeys in the world?

Response: Thank you for your suggestion and this is an important point to include. Changes have been made to include reference to both working or draft donkeys and donkeys used in production (milk, meat and skin).

Line 234: Diagnosis, not diagnose-

Response: Change has been made and it can be found on line 244.

Line 294: This is not clear-

Response: The study of the interaction between knowledge and gender may imply the multiplication of the number of possibilities within the interaction factor. Hence instead of considering two variables (knowledge and gender) with two (male and female) and three levels (minimal, intermediate and extensive knowledge), respectively, we would be considering one variable (knowledge level-gender) with six levels (male-minimal knowledge, female-minimal knowledge, male-intermediate knowledge, female-intermediate knowledge, male-extensive knowledge and female-extensive knowledge). Then, in the context of our sample and to maintain the balance in the representativity of each possibility, clustering both variables under an interaction term may reduce the frequency of cases across levels to two cases per level. This offers lower chances to compare than in a case in which the variables of knowledge level and gender are considered separately.

  1. Navas González, F.J.; León Jurado, J.M.; Delgado Bermejo, J.V. Interpretación de las interacciones entre variables categóricas en modelos de regresión categórica para el metaanálisis de efectos fijos de modelos animales. In Proceedings of XIX Simposio Iberoamericano sobre conservación y utilización de Recursos Zoogenéticos, Riobamba (Ecuador), 22nd to 26th October.

Line 358: there seems to be a relatively high number of false negatives and positives; this needs more discussion if readers are to be convinced of the claims made

Response: Now line 399-400

Line 415: should be 'were' not 'was'-

Response: Change has been made and can now be found on line 459.

Line 451: discussion of ethics and experimental controls. In principle, a control group receiving anesthesia without surgical intervention is possible. Please add a comment.

Response: Thank you for your comment and very true a control group that did not undergo the surgical procedure may have been possible but the study group owned by a rescue was scheduled to undergo castration independent of this study. Considering, observations and data was collected at several points prior to castration and administration of anesthesia and post-surgery, we believe the donkeys served as self-controls due to the fact that we were able to measure their changes in before pre and post surgery. Granted, you bring up a good point and we could have considered if all parties would have agreed to photograph the individual donkeys pre/post anesthesia and measure any differences seen in facial and body posture responses. The following statement has now been added to lines 528-33. Another option that may have been considered for a control would have been to have one group of donkeys undergo analgesic but no surgery procedure and another group complete the surgery. However, based on the fact other studies had used a similar protocol to this study we did not see it necessary nor in the best interest of the donkey’s welfare to place them under general anesthesia to simply photograph their face pre and post sedation.

Reviewer 2 Report

Overall comment

This article is generally well designed, and adds important data to the literature on donkey pain and behavior. While the article is focusing on donkeys kept as pets, it would be worthwhile mentioning the role of donkeys as working animals and the potential impact of the work on this population.

The study omits to discuss two recently published welfare assessment tools; the reviewer is not in any way affiliated with this work, but believes the assessment tools and the proposed grimace scale in this study could be of mutual benefit for welfare assessments:

Sommerville, R.; Brown, A.F.; Upjohn, M. A standardised equine-based welfare assessment tool used for six years in low and middle income countries. PLoS ONE 2018, 13, e0192354

Raw, Z.; Rodrigues, J.B.; Rickards, K.; Ryding, J.; Norris, S.L.; Judge, A.; Kubasiewicz, L.M.; Watson, T.L.; Little, H.; Hart, B.; Sullivan, R.; Garrett, C.; Burden, F.A. Equid Assessment, Research and Scoping (EARS): The Development and Implementation of a New Equid Welfare Assessment and Monitoring Tool. Animals 202010, 297.

Overall the paper would benefit from editing for improved style. The writing is hard to follow in places and there are numerous typographical errors. This paper should have been very easy to review, but carelessness in editing have made the process more lengthy than it should have been.

Detailed Review

Abstract and Summary

Line 22 – unlike

26 – ‘the disease’ – what specific disease? ‘disease’ may be better

31 – proved

38 and elsewhere throughout manuscript -  pre- and post-

45 – small p for p-values

47 – Can you present sensitivity, specificity and accuracy figures separately in the abstract? These are all of interest separately to the reader.

Introduction

Overall well written. Would be beneficial to discuss other populations of animals, eg, working and farmed donkeys, to which this research could apply

57 – few signs of?

75 – perspective instead of interest

Materials and Methods

133-134 – sentence doesn’t read quite right

146 - <4 hrs

160– More detail on sources used to develop the ethogram would be useful, as the basis for the current ethogram is unclear from the presented data

Figures 1- 5 – Text on arrows rather small and hard to read

Figure 1 legend suggests images are related to the pain scale, but don’t clarify in what way (e.g., which are intended to demonstrate more/less pain, if applicable). Re word or clarify.

209- Were photos presented to the observers as hard copies or in digital form? If in digital form, were viewing devices standardized, and were manipulations such as zooming in possible? Were all photos taken with the same camera? Please specify make/model of camera used.

234 – diagnosis

234/5 – What parameters did the veterinarian use to determine the presence or absence of pain?

270 – To clarify- the castrations were scheduled by the NGO, and were to be performed anyway regardless of the study?

Results

288 – small p for p-value

296-297 – Are the differences between males and females statistically significant? If so please cite p – value. Graph for fig 7 doesn’t make it very clear if these differences are significant or not.

298 – identify

Figures 6, 7, 8, 9- Text on images fuzzy, needs replacing

315 – ‘the figure represent’ – representative of? Represents? Unclear wording.

320 – text or test?

329 - signs

 330 - ‘the figure represent’ – representative of? Unclear wording.

Discussion

Generally good and well-referenced. Rather a lot of use of passive voice, try re wording in places as this will make the text easier to follow. For example, instead of saying “Several studies have confirmed that indicators of a painful/sick donkey may include sham eating, chewing, generalized dullness, shifting weight to contralateral limb, unresponsive/decreased mobility of ears, switching tail and tail tucked

[3-5]”, simply say ‘Indicators of a painful/sick donkey may include…. [refs].

There is also a mix of ‘veterinarian’ and ‘veterinary surgeon’ in some places. Both are acceptable but stick to one throughout.

386 – 388 – what behavioural responses were observed, were any comparable to or applicable to this study?

392-394 – Sentence doesn’t read well – re phrase

400 – Challenging to those? Not by those?

417 – Don’t need to specify the statistical test here, if presenting results should be in results

424 – the horse or horses

447 – Decision for

450 – is based upon

452 – the study/this study

453 – Sentence doesn’t really go anywhere

452 onwards – why use author name and cite reference for Jöchle?

466 – patent? Present?

452 – This paragraph is essential to explain some of the reasoning for the study, but it reads in a rather muddled way. The reviewer would suggest a re-write of this paragraph for clarity.

493 – p value for gender and supplementary information should be in results.

532 – drug

540 – Challenging by those or to those?

Author Response

Reviewer #2

Overall comment

This article is generally well designed, and adds important data to the literature on donkey pain and behavior. While the article is focusing on donkeys kept as pets, it would be worthwhile mentioning the role of donkeys as working animals and the potential impact of the work on this population.

The study omits to discuss two recently published welfare assessment tools; the reviewer is not in any way affiliated with this work, but believes the assessment tools and the proposed grimace scale in this study could be of mutual benefit for welfare assessments:

Sommerville, R.; Brown, A.F.; Upjohn, M. A standardised equine-based welfare assessment tool used for six years in low and middle income countries. PLoS ONE 2018, 13, e0192354

Raw, Z.; Rodrigues, J.B.; Rickards, K.; Ryding, J.; Norris, S.L.; Judge, A.; Kubasiewicz, L.M.; Watson, T.L.; Little, H.; Hart, B.; Sullivan, R.; Garrett, C.; Burden, F.A. Equid Assessment, Research and Scoping (EARS): The Development and Implementation of a New Equid Welfare Assessment and Monitoring Tool. Animals 202010, 297.

Response: Discussion added “The original lack of scientific support in regards to the assessment of specific donkey welfare contrasts the advances that have recently appeared in the scene. This methodological revolution appeared as an effort to address the misattributions and misconceptions of the donkey species. Incorrect evaluation of donkey welfare often bases on an incorrect perception of the behavioural, physiological, pathological idiosyncrasies of the species, among others. In this context, complex validated welfare approaches [1] can benefit from the methodologies implemented by large scale applicable methods [2] to report replicable and still trustable results in contexts such as medium or low income countries, for which donkey is a common and relatively frequent element. The present research can complement the aforementioned methods as, it focuses on the human ability to determine signs of pain, which according to literature has often been deemed difficult provided to subtleness of donkeys to display signs of distress [3].”

Overall the paper would benefit from editing for improved style. The writing is hard to follow in places and there are numerous typographical errors. This paper should have been very easy to review, but carelessness in editing have made the process more lengthy than it should have been.

Response: We apologize and tried to address all the comments and suggestions made by reviewers.

Detailed Review

Abstract and Summary

Line 22 – unlike-

Response: Change has been made

26 – ‘the disease’ – what specific disease? ‘disease’ may be better-

Response: “the” has been removed

31 – proved-

Response: Change has been made to “proved”

38 and elsewhere throughout manuscript -  pre- and post-

Response: Change has been made to “pre-“ and “post-“

45 – small p for p-values

Response: Change has been made to “p”

47 – Can you present sensitivity, specificity and accuracy figures separately in the abstract? These are all of interest separately to the reader.

Response: Separate values were reported in the abstract.

Introduction

Overall well written. Would be beneficial to discuss other populations of animals, eg, working and farmed donkeys, to which this research could apply

Response: Additional information was provided in regards other equine populations, specifically highlighting differences between horses and donkeys.

57 – few signs of-

Response: Change has been made and is now on line 60

75 – perspective instead of interest-

Response: Change has been made and can be found on line 80

Materials and Methods

133-134 – sentence doesn’t read quite right-

Response: Sentence has been changed and now reads: A modification of the "Miller's" knot, was used for transfixation and ligation of the spermatic cords. The ligation process was completed by using Coated Polyglactin 910 and Triclosan suture (Coated Vicryl Plus, Ethicon, Somerville, NJ 08876, USA).

146 - <4 hrs

Response: Change has been made

160– More detail on sources used to develop the ethogram would be useful, as the basis for the current ethogram is unclear from the presented data-

Response: Sources have now been referenced: 10, 11, and 15 for the development of the ethogram.

Figures 1- 5 – Text on arrows rather small and hard to read

Response: Text on arrows was enlarged as possible. Anyway it reduces when the image is placed in the document.

Figure 1 legend suggests images are related to the pain scale, but don’t clarify in what way (e.g., which are intended to demonstrate more/less pain, if applicable). Re word or clarify.- Response: Changed to “Ethogram describing ear position related to the scale (0- not present, 1-moderately present, and 2- obviously present). Painful positions maybe associated with both ears back (C ), down (D), one ear forward, one to the side (E ), One ear to the side and one back (F), one ear to the side and one down (H), and one forward and one down (I) along with other facial action units associated with pain. “

209- Were photos presented to the observers as hard copies or in digital form? If in digital form, were viewing devices standardized, and were manipulations such as zooming in possible? Were all photos taken with the same camera? Please specify make/model of camera used.

Response: Change and information has been included: The survey included 54 photographs taken with an Apple mini ipad (version iso 11.4.1), six pictures per donkey: lateral, frontal head, and lateral body view (one pre and post castration photograph/view). Each observer took the survey once per day on their own device (e.g. computer, laptop, tablet or smart phone), for three consecutive days to ensure intraobserver reliability. Photographs were presented in a digital form and survey to the observers in the same order on each consecutive day.

234 – diagnosis-

Response: Change has been made, now found on line 258

234/5 – What parameters did the veterinarian use to determine the presence or absence of pain?

Response: The evaluation of pain based on the evidences reported by Dyson, et al. [4], who suggested the recognition of certain behavioral features may act as potential indicators of musculoskeletal pain, which may enable the early recognition of certain distressing conditions in equids. The indicators proposed by Gleerup and Lindegaard [5] were considered as behavioral indicators of pain in the context of normal equine behavior. The donkeys were physiologically evaluated by veterinarians before their free-from-pain condition could be ensured and these could be returned to their habitual housing. The review by Ashley, et al. [6] was considered to adjust the information and protocols of identification of pain related signs to the specific behavioral nature of donkeys.

270 – To clarify- the castrations were scheduled by the NGO, and were to be performed anyway regardless of the study?

Response: Yes, the castrations were scheduled by the NGO regardless if we were observing or not. This has been clarified in the sentence. Castrations were scheduled and performed by a veterinary nongovernmental organization on-farm (in the field) independent of the study.

Results

288 – small p for p-value-

Response: Change has been made and can now be found on line 315

296-297 – Are the differences between males and females statistically significant? If so please cite p – value. Graph for fig 7 doesn’t make it very clear if these differences are significant or not.-

Response: Yes, there’s a statistical significance, the p-value has been added, p-value = 0.001 and added to figure 7 (now line 358)

298 – identify-

Response: Change has been made now line 338

Figures 6, 7, 8, 9- Text on images fuzzy, needs replacing

Response: Image quality was improved.

315 – ‘the figure represent’ – representative of? Represents? Unclear wording.

Response: Change has been made and now reads, “Frequencies of pain identification success possibilities across the different levels of donkey knowledge from minimal, moderate to extensive in both pre-and post-castration images.”

320 – text or test?

Response: Change has been made to “test” from “text” now line 359

329 – signs-

Response: Change has been made and now all places read and are spelled “signs” now line 374

 330 - ‘the figure represent’ – representative of? Unclear wording.

Response: Change has been made and sentence deleted for clarification.

Discussion

Generally good and well-referenced. Rather a lot of use of passive voice, try re wording in places as this will make the text easier to follow. For example, instead of saying “Several studies have confirmed that indicators of a painful/sick donkey may include sham eating, chewing, generalized dullness, shifting weight to contralateral limb, unresponsive/decreased mobility of ears, switching tail and tail tucked

[3-5]”, simply say ‘Indicators of a painful/sick donkey may include…. [refs].

Response: Thank you for your suggestion and the changes have been included. The sentence now reads, “In previous studies, indicators of a donkey in pain or sick may include the following behavioral signs associated with pain such as a slight twitch to the tail, but authors claimed they were subtler than those of horse [6-11].”

There is also a mix of ‘veterinarian’ and ‘veterinary surgeon’ in some places. Both are acceptable but stick to one throughout.

Response: We changed it.

386 – 388 – what behavioural responses were observed, were any comparable to or applicable to this study? Change has been made and includes the following wording-

Response: “until this study was conducted [8]. The horse study assessed pain in similar areas to this study such as orbital tightening, muzzle and ear position and similar indicators were observed in donkeys in this study in the same facial areas. “ now lines 441-443.

392-394 – Sentence doesn’t read well – re phrase.

Response: Change has been made and now reads, “Granted, pain behavior may be dependent upon individual animal basis and signs of discomfort may vary depending on the type of pain if it’s acute versus chronic.” Now lines 450-451.

400 – Challenging to those? Not by those?

Response: Change has been made to “to those” and now on line 454

417 – Don’t need to specify the statistical test here, if presenting results should be in results Change has been made and sentence now reads,

Response: “One of the most accurate body language markers we found for detecting distress related to pain was the overall appearance of the donkey and body stance. Our study’s results indicate the importance of observing more than just the face of the donkey to identify signs of pain.” Lines 474-76.

424 – the horse or horses-

Response: Change has been made to “horses” and now line 481

447 – Decision for-

Response: Change has been made to “for” and now line 522

450 – is based upon-

Response: Change has been made to “is based upon” and now line 525

452 – the study/this study-

Response: Change has been made to “this” and is now line 527

453 – Sentence doesn’t really go anywhere- Sentence has been re-worded and now reads,

Response: “Jöchle [12] like this study used donkeys as self-controls when testing for the palliative effect of detomidine as a sedative and analgesic agent in cases of severe pain (abdominal colic).”

452 onwards – why use author name and cite reference for Jöchle? Change has been made and sentence now reads,

Response: “Donkeys have been used as self-controls in other studies such as one study that tested for the palliative effect of detomidine as a sedative and analgesic agent in cases of severe pain during abdominal colic [32].”

466 – patent? Present?-

Response: Change has been made to “present” now line 541

452 – This paragraph is essential to explain some of the reasoning for the study, but it reads in a rather muddled way. The reviewer would suggest a re-write of this paragraph for clarity.

Response: Change has been made (line 527-552) and now reads:

The second main reason this study was designed using donkeys as their own self-control was due to findings in previous donkey studies attempting to measure and define pain behavior and measure clearance of analgesic agents. Donkeys were used as self-controls in a study that tested for the palliative effect of detomidine as a sedative and analgesic in cases where severe pain was reported during abdominal colic [32]. The severity of abdominal pain accompanying colic was determined prior to (time 0) and after drug administration (15, 30, 45 and 60 minutes) using a standardized scoring system to describe several clinical parameters common to equine colic. Among these clinical parameters, the presence or absence of body signs was considered using a scale from no evidence of signs of discomfort to severe evidence of signs of discomfort. This study similar to our study photographed the donkeys as various times [32]. For our study, additional times were set to take photos prior to castration, as a way to monitor the animals to make sure no evident changes occurred prior to castration at 48, 24, 0 hours before surgery. Amin and Najim [13] also compared detomidine, ketamine and other anesthetics stated full recovery was achieved at 40 min in donkeys. In regards to the metabolism of analgesics, donkeys may describe two different trends, one characterizing the period of time that takes for half-life to be reached, then afterward the period of time needed for the analgesic to disappear from the body or at least its effects to not be present. Contextually, Grosenbaugh, et al. [14], reported NSAIDs may be clinically effective longer in donkeys than in horses as indicated by the respective plasma t1/2 (half time in donkeys ranges from 0.75 to 4.5 hours, with maximum limit only being half an hour longer than in horses). The study suggested a faster reduction of drug plasma levels after the drug had reached half its concentration (half-life) provided the notably increased clearance times of 1.78 (ml/kg bwt/h), which almost double those in horses [34]. Bearing this in mind and to ensure the effects of any of the drugs did not alter the donkeys’ behavior, post castration photos were taken 8, 24- and 48-hours post-surgery. Another comparison to this study was the differences in scales, our study developed a scale that included a “don’t know” possibility to be able to gather those cases in which observers were not able to identify the degree of expression of a certain body sign related to pain which was not included in other studies [32,34].

493 – p value for gender and supplementary information should be in results.

Response: Change has been made and the p-value has been removed (now line 640)

532 – drug-

Response: Change has been made (now line 681)

540 – Challenging by those or to those?

Response: Change has been made (now line 689).